# A nonparametric alternative to the Cochran-Armitage trend test in genetic case-control association studies: The Jonckheere-Terpstra trend test

Sydney E. Manning[1], Hung-Chih Ku[2], Douglas F. Dluzen[3], Chao Xing[4]*, Zhengyang Zhou[5]*

1 Department of Pharmacotherapy, University of North Texas Health Science Center, Fort Worth, TX, United States of America, 2 Department of Mathematical Sciences, DePaul University, Chicago, IL, United States of America, 3 Department of Biology, Morgan State University, Baltimore, Maryland, MD, United States of America, 4 McDermott Center for Human Growth and Development and Department of Bioinformatics, University of Texas Southwestern Medical Center, Dallas, TX, United States of America, 5 Department of Biostatistics and Epidemiology, University of North Texas Health Science Center, Fort Worth, TX, United States of America

* zhengyang.zhou@unthsc.edu (ZZ); chao.xing@utsouthwestern.edu (CX)

**Data Availability Statement:** Real data analyzed in this study can be obtained from the cited articles

## Abstract

Identifications of novel genetic signals conferring susceptibility to human complex diseases is pivotal to the disease diagnosis, prevention, and treatment. Genetic association study is a powerful tool to discover candidate genetic signals that contribute to diseases, through statistical tests for correlation between the disease status and genetic variations in study samples. In such studies with a case-control design, a standard practice is to perform the Cochran-Armitage (CA) trend test under an additive genetic model, which suffers from power loss when the model assumption is wrong. The Jonckheere-Terpstra (JT) trend test is an alternative method to evaluate association in a nonparametric way. This study compares the power of the JT trend test and the CA trend test in various scenarios, including different sample sizes (200–2000), minor allele frequencies (0.05–0.4), and underlying modes of inheritance (dominant genetic model to recessive genetic model). By simulation and real data analysis, it is shown that in general the JT trend test has higher, similar, and lower power than the CA trend test when the underlying mode of inheritance is dominant, additive, and recessive, respectively; when the sample size is small and the minor allele frequency is low, the JT trend test outperforms the CA trend test across the spectrum of genetic models. In sum, the JT trend test is a valuable alternative to the CA trend test under certain circumstances with higher statistical power, which could lead to better detection of genetic signals to human diseases and finer dissection of their genetic architecture.

## Introduction

Over the past fifteen years, genome-wide association studies have significantly expanded the knowledge base for genetic factors in important healthcare outcomes [1]. Such studies have

(Liu et al., 2000; Timmann et al., 2012; Loley et al., 2013).

**Funding:** This work is supported by the National Institute of Environmental Health Sciences grant R03ES034138 to C.X. and Z.Z. D.F.D. and Z.Z. are also supported in part by the National Institute on Minority Health and Health Disparities grant 5U54MD013376-8281. The content is solely the responsibility of the authors and does not necessarily represent the official views of the National Institutes of Health. The funders had no role in study design, data collection and analysis, decision to publish, or preparation of the manuscript. There was no additional external funding received for this study.

**Competing interests:** The authors have declared that no competing interests exist.

identified numerous genetic signals contributing to various complex human diseases, which can be very important to the diseases' diagnosis, prevention, and treatment. One commonly used approach to test association in a case-control genetic study is the Cochran-Armitage (CA) trend test [2, 3] under the assumption of an additive genetic model [4, 5], which can reach the optimal power when the underlying genetic model is also additive. However, it can suffer from power loss when the true genetic model is nonadditive (see, e.g., [6–9]). Power loss is one critical issue in genetic association studies. On the one hand, conducting a statistical test with reduced power may fail to detect true genetic signals, leading to false negative results. On the other hand, to achieve the same level of statistical power, the sample sizes needed will increase, leading to higher study expenses and resource requirements.

To test for associations between the disease status and genetic variation, one alternative approach to the CA trend test is the Jonckheere-Terpstra (JT) trend test [10, 11], which is a rank-based nonparametric test. The JT trend test does not make assumptions on genetic models or data distribution, and thus has the potential to achieve better statistical power than the parametric CA trend test under certain circumstances. The potentially higher power from the JT trend test may result in novel genetic discoveries for complex human diseases, which can help researchers better understand the genetic etiology and eventually aid in the development of effective diagnosis, prevention, and treatment strategies of the diseases. Although the JT trend test offers an alternative to the CA trend test with potential advantages, their comparative performance has not been examined in the genetic literature. In this study, we aim to fill this research gap by comparing the power of the two tests in various conditions via simulations and real data analysis. The knowledge gained in this study can help guide the model selection between the CA and JT trend tests when conducting genetic case-control studies in practice.

## Methods

Consider a diallelic locus with the major and minor alleles denoted as $a$ and $A$, respectively, the genotype distribution in a case-control study can be summarized as in Table 1. Specifically, denote $r_i$ and $s_i$ as the number of cases and controls, respectively, for genotype $G_i$, where $i \in \{0,1,2\}$ reflects the number of $A$ alleles a subject has. Thus $G_0$, $G_1$, and $G_2$ correspond to genotypes $aa$, $Aa$, and $AA$, respectively. Denote by $R$, $S$, and $n_i$ the marginal sums such that $R = \sum_{i=0}^{2} r_i$, $S = \sum_{i=0}^{2} s_i$, and $n_i = r_i + s_i$, and by $N$ the total sample size such that $N = R + S = \sum_{i=0}^{2} n_i$. Assume $(r_0, r_1, r_2)$ follow a trinomial distribution with parameters $R$ and $(\tau_0, \tau_1, \tau_2)$, and $(s_0, s_1, s_2)$ follow a trinomial distribution with parameters $S$ and $(v_0, v_1, v_2)$. The null hypothesis of no association between the disease and genotype is then $H_0$: $\tau_i = v_i$, for $i \in \{0,1,2\}$. Equivalently, we can also assume $r_i$'s are drawn from binomial distributions $Bin(n_i, \pi_i)$. The null hypothesis of no association between the disease and genotype is $H_0$: $\pi_0 = \pi_1 = \pi_2$. Assuming $G_0$, $G_1$, and $G_2$ are three ordered categories, a restricted alternative hypothesis for a trend test is $H_1$: $\pi_0 \leq \pi_1 \leq \pi_2$ or $\pi_0 \geq \pi_1 \geq \pi_2$ with at least one strict inequality.

**Table 1. Genotype distribution at a diallelic marker in a case-control study.**

| Phenotype | Genotype* | | | Total |
|---|---|---|---|---|
| | $aa$ | $Aa$ | $AA$ | |
| Cases | $r_0$ | $r_1$ | $r_2$ | $R$ |
| Controls | $s_0$ | $s_1$ | $s_2$ | $S$ |
| Total | $n_0$ | $n_1$ | $n_2$ | $N$ |

* $A$ denotes the minor allele across the paper.

To test $H_1$, the CA trend test assigns a set of scores $(x_0, x_1, x_2)$ to $G_0$, $G_1$, and $G_2$, respectively, with the constraints $x_0 \leq x_1 \leq x_2$ and $x_0 < x_2$, and examines whether there is a linear relationship between $\pi_i$'s and $x_i$'s by fitting a linear regression model. The test statistic is

$$T_{CA} = \frac{N(N\sum_{i=0}^{2} r_i x_i - R\sum_{i=0}^{2} n_i x_i)^2}{RS[N\sum_{i=0}^{2} n_i x_i^2 - (\sum_{i=0}^{2} x_i n_i)^2]}.$$ Under $H_0$, $T_{CA}$ follows a $\chi^2$ distribution with 1 degree-of-freedom (d.f.). The choices of $(x_0, x_1, x_2)$ represent assumptions on the genetic models. In practice, the additive model with $(x_0, x_1, x_2) = (0, 0.5, 1)$ is usually assumed because of its robustness. Hereinafter we denote it as $T_{CA}^{Add}$.

Alternatively, the JT trend test compares the ranks of subjects based on their affection status between genotype groups to test $H_1$. Consider the disease status of case and control as an ordinal variable $Y$, and denote by $Y_{ij} \in \{0,1\}$ individual $j$'s phenotypic value with genotype $G_i$. The JT test statistic is $T_{JT} = \frac{[U-E(U)]^2}{Var(U)}$, where $U = \sum_{j=1}^{n_0} \sum_{k=1}^{n_1} S(Y_{0j}, Y_{1k}) + \sum_{j=1}^{n_0} \sum_{k=1}^{n_2} S(Y_{0j}, Y_{2k}) +$ $\sum_{j=1}^{n_1} \sum_{k=1}^{n_2} S(Y_{1j}, Y_{2k})$, $E(U) = \frac{N^2 - \sum_{i=0}^{2} n_i^2}{4}$, and $Var(U) = \frac{A}{72} + \frac{B}{36N(N-1)(N-2)} + \frac{C}{8N(N-1)}$, in which $A = N(N-1)(2N+5) - \sum_{i=0}^{2} n_i(n_i-1)(2n_i+5) - S(S-1)(2S+5) - R(R-1) \times$ $(2R+5)$, $B = \sum_{i=0}^{2} n_i(n_i-1)(n_i-2)[S(S-1)(S-2) + R(R-1)(R-2)]$ and $C = \sum_{i=0}^{2} n_i(n_i-1)[S(S-1) + R(R-1)]$. The components of $U$ are the Mann-Whitney U statistics defined as $S(Y_{\cdot j}, Y_{\cdot k}) = \begin{cases} 1, \text{if } Y_{\cdot j} < Y_{\cdot k} \\ 0.5, \text{if } Y_{\cdot j} = Y_{\cdot k} \\ 0, \text{if } Y_{\cdot j} > Y_{\cdot k} \end{cases}$. For large N and $n_i$'s not too small, $T_{JT}$ also follows a $\chi^2$ distribution with 1 d.f. under $H_0$.

For simplicity, hereinafter we use $T_{JT}$ and $T_{CA}^{Add}$ to refer to both tests and test statistics.

## Simulation

We conduct simulations to compare performance between $T_{JT}$ and $T_{CA}^{Add}$ in terms of statistical power under various conditions. Define the penetrance function for each genotype as $f_i = P$(affected$|G_i$), $i = 0, 1, 2$, and define genotype relative risk as $\lambda_i = f_i/f_0$; thus $\lambda_0 = 1$. The dominant, additive, and recessive genetic models can be specified by $\lambda_1 = \lambda_2$, $\lambda_1 = (1+\lambda_2)/2$, and $\lambda_1 = 1$, respectively. Note that the genetic model is defined in regard to the minor allele in this study. The model can be reparameterized by defining $\lambda_1 = 1+\lambda cos\theta$ and $\lambda_2 = 1+\lambda sin\theta$, where $\lambda \geq 0$ is the distance between point $P = (\lambda_1, \lambda_2)$ and point $O = (1,1)$, which determines how far the genetic effect is from the null, and $\theta \in [\pi/4, \pi/2]$ is the angle between $OP$ and the horizontal line in a two-dimensional space, which determines the genetic model [12]. Therefore, the null hypothesis can be rewritten as $H_0: \lambda = 0$. In terms of genetic models, $\theta = \pi/4$, arctan 2, and $\pi/2$ correspond to dominant, additive, and recessive models, respectively. We performed simulations under the following alternative settings. Assume a disease prevalence ($K$) of 0.1 and the minor allele frequency (MAF) $q \in (0.05, 0.1, 0.2, 0.3)$. Fix the alternative hypothesis as $\lambda = 1$ and vary the genetic models by setting $\theta^* = \theta/\pi$ from 1/4 to 1/2, i.e., from the dominant model to the recessive model, with an increment of 0.01. Under each genetic model, penetrances are determined by $f_0 = K/[(1-q)^2 + 2\lambda_1 q(1-q) + \lambda_2 q^2]$ and $f_i = \lambda_i f_0$. The probabilities of the two trinomial distributions for cases and controls are then $\tau_i = P(G_i)f_i/K$ and $v_i = P(G_i)(1-f_i)/(1-K)$, respectively. Consider a balanced design, i.e., $R = S$ with the total sample size $N \in (200, 500, 1000)$. At each setting 10,000 replicates are simulated, and each dataset is examined by both $T_{JT}$ and $T_{CA}^{Add}$. The empirical power is estimated as the proportion of the replicates for which the p-value is less than or equal to 0.05. In an additional set of simulations for wider range of sample size and MAF, we considered $N = 1500$ and 2000 as well as $q = 0.4$ across the genetic models.

Because the sample sizes and MAF were large, the effect size in the alternative hypothesis was set to $\lambda = 0.5$ to make the maximum power less than 1 for the sake of comparison.

The simulation results across the main settings are presented in Fig 1 and the results for the additional set of simulations are presented in S1 Table. In all situations $T_{JT}$ is more powerful

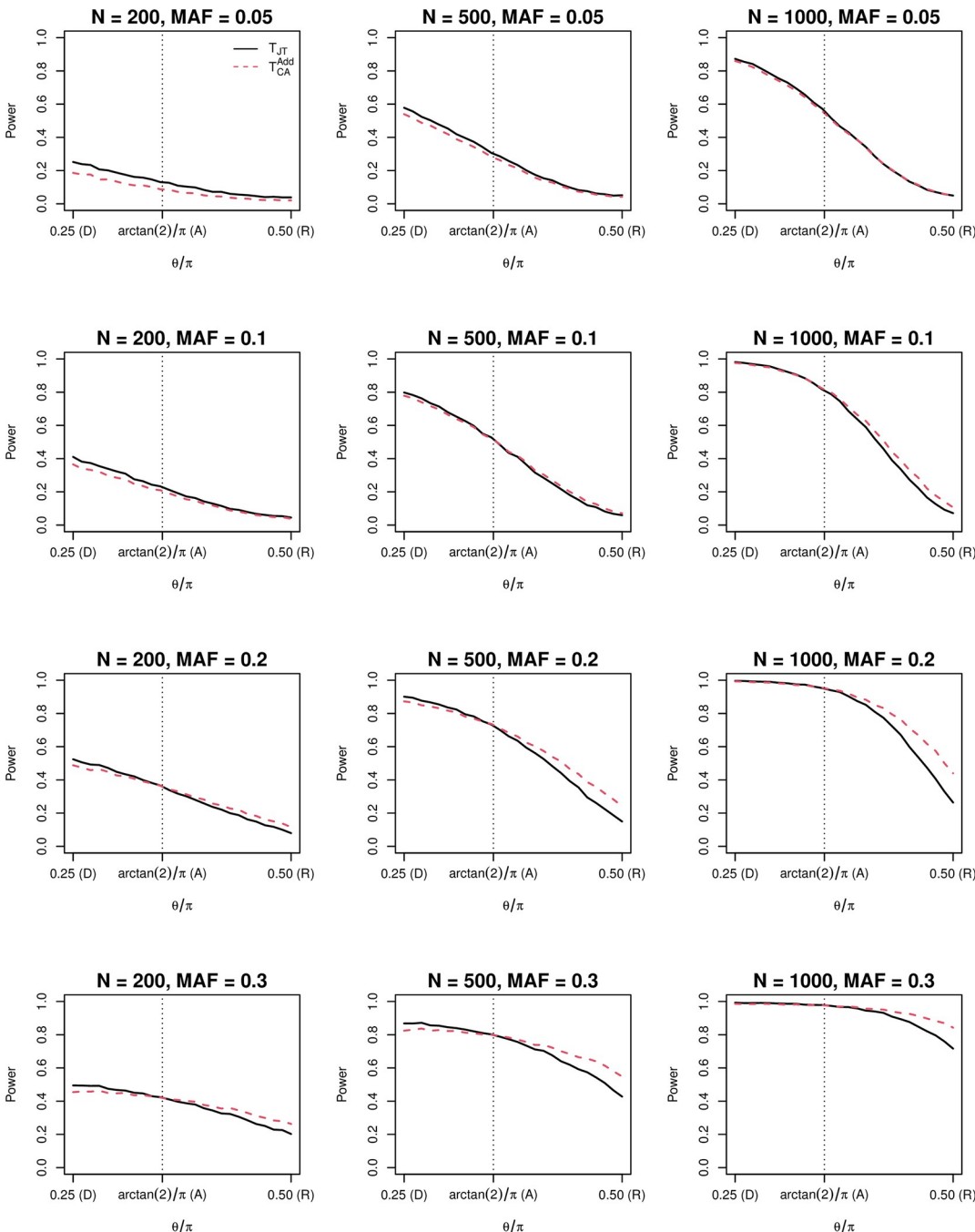

**Fig 1. Power comparison between the Jonckheere-Terpstra trend test ($T_{JT}$) and the Cochran–Armitage trend test ($T_{CA}^{Add}$).** The black solid line denotes $T_{JT}$ and the red dashed line denotes $T_{CA}^{Add}$. Along the x-axis, $\theta = \pi/4$, arctan 2, and $\pi/2$ correspond to dominant (D), additive (A), and recessive (R) genetic models, respectively. The y-axis is the average empirical power at the 0.05 level based on 10,000 replicates each. The disease prevalence equals 0.1. MAF: minor allele frequency.

than $T_{CA}^{Add}$ when the underlying genetic model is dominant. The power advantage of $T_{JT}$ diminishes as the genetic model evolves toward the additive model. In most situations except for small sample sizes and low MAFs, the two tests have approximately equivalent power when the underlying model is additive. $T_{JT}$ becomes less powerful than $T_{CA}^{Add}$ and the disadvantage enlarges when the genetic model keeps evolving toward the recessive end. In case of low MAFs and small sample sizes, e.g., $N = 200$ & $q \leq 0.1$ or $N \leq 1000$ & $q = 0.05$, $T_{JT}$ is more powerful than $T_{CA}^{Add}$.

To verify the above findings, for each simulation setting we construct a table with the value in each cell equal to the expected value under the trinomial distributions for cases and controls, respectively, i.e., $E(r_i) = \tau_i N/2$ and $E(s_i) = v_i N/2$, $i \in \{0,1,2\}$. Specifically, in each simulation setting, using the fixed combination of sample size ($N$), MAF ($q$), and genetic model ($\theta$), the cell probabilities of each genotype for cases and controls in the genotype distribution table (Table 1) can be calculated, and therefore, the expected cell values can also be calculated by multiplying the probabilities with the sample size. This table consists of the expected cell values, which allows us to evaluate the relative performance of the two tests by comparing their theoretical test statistics ($T_{JT}$ and $T_{CA}^{Add}$) in each simulation setting. For each table, the theoretical $T_{JT}$ and $T_{CA}^{Add}$ are calculated and compared by $\Delta T = \frac{(T_{JT} - T_{CA}^{Add})}{T_{CA}^{Add}} \times 100\%$. Therefore, a positive (or negative) value of $\Delta T$ indicates the JT trend test is more (or less) powerful than the CA trend test. The results of $\Delta T$ for dominant, additive, and recessive genetic models across simulation settings are reported in Table 2. Consistent with the simulation results, $\Delta T > 0$ when the genetic model was dominant; $|\Delta T| < 2\%$ when the genetic model was additive, and $\Delta T < 0$ when the genetic model was recessive. The only discrepancy is that in case of low MAFs and small sample sizes, theoretically $\Delta T$ would be less than zero under the recessive model, but empirically it is greater than zero. We suspect it is because the parametric assumptions and asymptotic theory behind $T_{CA}^{Add}$ do not hold in these circumstances, whereas $T_{JT}$ does not impose assumptions on data distribution.

## Real data analysis

We further compared $T_{JT}$ and $T_{CA}^{Add}$ in real data, which confirmed the simulation results. The first example is on the association between the variant rs2398162 and hypertension in the

**Table 2. Percent difference between the Jonckheere-Terpstra trend test statistic ($T_{JT}$) and the Cochran-Armitage trend test statistic ($T_{CA}^{Add}$) based on the expectation of genotype distributions under dominant, additive, and recessive genetic models.**

| N | Minor Allele Frequency | $(T_{JT} - T_{CA}^{Add})/T_{CA}^{Add}$ | | |
|---|---|---|---|---|
| | | Dominant | Additive | Recessive |
| 200 | 0.05 | 2.15% | -1.39% | -69.47% |
| 200 | 0.1 | 4.97% | -1.67% | -62.83% |
| 200 | 0.2 | 9.92% | -1.00% | -46.62% |
| 200 | 0.3 | 11.58% | 0.14% | -27.71% |
| 500 | 0.05 | 2.46% | -1.10% | -69.38% |
| 500 | 0.1 | 5.28% | -1.38% | -62.71% |
| 500 | 0.2 | 10.25% | -0.70% | -46.46% |
| 500 | 0.3 | 11.91% | 0.44% | -27.49% |
| 1000 | 0.05 | 2.56% | -1.00% | -69.35% |
| 1000 | 0.1 | 5.39% | -1.28% | -62.68% |
| 1000 | 0.2 | 10.36% | -0.60% | -46.41% |
| 1000 | 0.3 | 12.03% | 0.54% | -27.42% |

Wellcome Trust Case Control Consortium study [13]. There were 1940 cases ($r_0 = 1205$, $r_1 = 624$, $r_2 = 111$) and 2923 controls ($s_0 = 1608$, $s_1 = 1121$, $s_2 = 194$), and it was suggested the minor allele have a dominant effect. Applying the two trend tests on the dataset, we can obtain $T_{JT} = 22.82$ (p-value = $1.8 \times 10^{-6}$) and $T_{CA}^{Add} = 19.97$ (p-value = $7.9 \times 10^{-6}$). These results were in line with the observations in the simulation that $T_{JT}$ is more powerful than $T_{CA}^{Add}$ when the underlying genetic model is dominant.

Additional real data analyses came from case-control studies for falciparum malaria [14], age-related macular degeneration (AMD) [15] and hypertension with additional variants, with the genotype counts all extracted from [9]. In the falciparum malaria study, variant rs10900589 in *ATP2B4* was associated with the disease in the Ghanaian samples. This association was also evaluated in the Gambian samples and it was significant under a recessive model but insignificant under dominant and additive models. In the AMD study, variants rs380390 and rs10131337 in *CFH* were associated with AMD. We examined the associations of the three variants with the diseases using both tests. Cell counts, test statistics and p-values are reported in Fig 2. For rs10900589, the minor allele approximately acts in a recessive mode $\left( \frac{r_0}{n_0} \approx \frac{r_1}{n_1} < \frac{r_2}{n_2} \right)$ and $T_{JT} < T_{CA}^{Add}$, which were consistent with the simulation results that $T_{JT}$ is less powerful than $T_{CA}^{Add}$ when the underlying genetic model is recessive. For both rs380390 and rs10131337, the minor allele approximately acts in an additive mode $\left( \frac{r_1}{n_1} \approx \frac{r_0/n_0 + r_2/n_2}{2} \right)$ and $T_{JT} \approx T_{CA}^{Add}$, which were consistent with the simulation results that the two tests have comparable power when the underlying model is additive. In the hypertension study, we compared the two tests on three SNPs that showed genome-wide significance. The results are reported in S2 Table. The conclusion still holds in this real data analysis: $T_{JT}$ and $T_{CA}^{Add}$ had similar power when the genetic model was close to be additive (rs7961152, rs1937506, rs6997709).

To assess the potential genotyping errors among the variants considered in the real data analysis, we tested the Hardy-Weinberg equilibrium (HWE) among the cases and controls, separately, for each variant [16, 17]. An exact test for HWE was conducted using the R package HardyWeinberg [18], and the results with exact p-values for the variants were reported in S3 Table. Results showed that the p-values of the HWE tests for all the variants were larger than 0.01, with only two between 0.01 and 0.05, suggesting that there was little evidence of genotyping error among the variants. Moreover, we conducted allelic test to evaluate the association for the variants. The allelic test assesses the genetic association at the allele level by collapsing the genotypes into the counts for reference and alternative alleles, between cases and controls, however, this approach is not robust against the HWE departures [4]. The test statistics and p-values of allelic test results were summarized in S3 Table. Of note, the results were close to those of $T_{CA}^{Add}$, suggesting that the assumptions of HWE were not violated.

## Discussion

Our previous work elucidates the mechanism of the CA trend test that it examines the location shift of genotype scores between the case and control groups [19] by measuring the goodness of fit of a linear regression model correlating proportions of cases in genotype groups with their respective scores [20]. The preassigned scores reflect assumptions on the genetic model. In contrast, the JT trend test examines the location shift of phenotype scores among genotype groups in a rank-based nonparametric way without making assumptions on genetic models or data distribution. The power difference between the two tests in different situations shown in this study can be explained by their properties. When the underlying model is dominant, $T_{CA}^{Add}$ suffers power loss because of the wrong model assumption that it is inferior to $T_{JT}$; when the underlying model is recessive, the limited information on location shift hampers $T_{JT}$ more

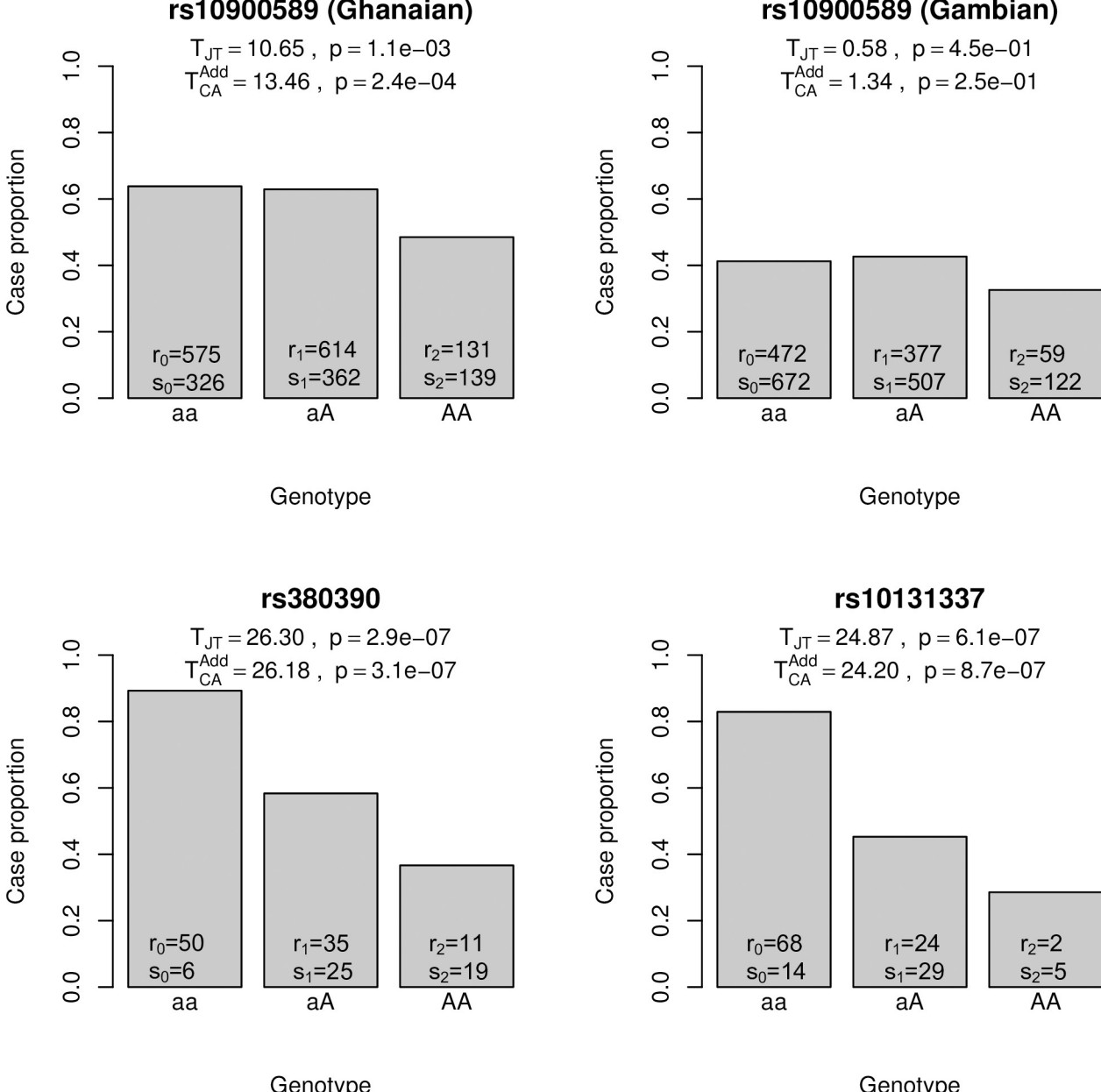

**Fig 2. Comparison between the Jonckheere-Terpstra trend test ($T_{JT}$) and the Cochran–Armitage trend test ($T_{CA}^{Add}$) in four real datasets.** *A* denotes the minor allele; $r_i$'s and $s_i$'s are as defined in Table 1.

than the wrong model assumption hampers $T_{CA}^{Add}$; in case of low MAFs and small sample sizes where the large-sample theory breaks, $T_{JT}$ outperforms $T_{CA}^{Add}$ because it does not impose assumptions on data distribution as the latter does.

In sum, in this study we compared the power of $T_{CA}^{Add}$ and $T_{JT}$ under different situations. By simulation and real data examples, we show $T_{JT}$ can provide a valuable alternative to $T_{CA}^{Add}$ in case of small sample sizes and low MAFs; when the genetic mechanism is known to be dominant, or that is the only model of interest, $T_{JT}$ is preferred. However, in a moderate to large sample size study with the true mode of inheritance unknown, the use of the JT trend test is

not recommended compared to the CA trend test under an additive model, which is more robust under a wide range of modes of inheritance.

## Supporting information

**S1 Table. Power comparison between $T_{JT}$ and $T_{CA}^{Add}$ for the additional simulation settings ($q$ = 0.4).**
(DOCX)

**S2 Table. Comparison between the Jonckheere-Terpstra trend test ($T_{JT}$) and the Cochran-Armitage trend test ($T_{CA}^{Add}$) on SNPs that were reported associated with hypertension.**
(DOCX)

**S3 Table. Exact p-values of the Hardy-Weinberg equilibrium (HWE) tests among cases and controls of the variants and the allelic test statistics in the Real data analysis.**
(DOCX)

## Acknowledgments

The authors acknowledge the Texas Advanced Computing Center (https://www.tacc.utexas.edu) at The University of Texas at Austin for providing high performance computing resources that have contributed to the research results reported within this paper.

## Author Contributions

**Conceptualization:** Hung-Chih Ku, Chao Xing, Zhengyang Zhou.

**Data curation:** Sydney E. Manning, Chao Xing, Zhengyang Zhou.

**Formal analysis:** Sydney E. Manning, Chao Xing, Zhengyang Zhou.

**Funding acquisition:** Zhengyang Zhou.

**Investigation:** Sydney E. Manning, Hung-Chih Ku, Douglas F. Dluzen, Chao Xing, Zhengyang Zhou.

**Methodology:** Hung-Chih Ku, Chao Xing, Zhengyang Zhou.

**Project administration:** Chao Xing, Zhengyang Zhou.

**Supervision:** Chao Xing, Zhengyang Zhou.

**Validation:** Sydney E. Manning, Douglas F. Dluzen, Chao Xing, Zhengyang Zhou.

**Visualization:** Sydney E. Manning, Chao Xing, Zhengyang Zhou.

**Writing – original draft:** Sydney E. Manning, Douglas F. Dluzen, Chao Xing, Zhengyang Zhou.

**Writing – review & editing:** Sydney E. Manning, Douglas F. Dluzen, Chao Xing, Zhengyang Zhou.

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
