## [Decision Letter · Decision Letter 0]

1 Mar 2022

PONE-D-21-39215A nonparametric alternative to the Cochran-Armitage trend test in genetic case-control association studies: the Jonckheere-Terpstra trend testPLOS ONE

Dear Dr. Zhou,

Thank you for submitting your manuscript to PLOS ONE. After careful consideration, we feel that it has merit but does not fully meet PLOS ONE’s publication criteria as it currently stands. Therefore, we invite you to submit a revised version of the manuscript that addresses the points raised during the review process.

We look forward to receiving your revised manuscript.

Kind regards,

Mehdi Rahimi, Ph.D.

Academic Editor

PLOS ONE

Journal Requirements:

2.Thank you for stating in your Funding Statement: 

(This work was supported by National Institute on Minority Health and Health Disparities 5U54MD013376-8281 (ZZ). 

The funders had no role in study design, data collection and analysis, decision to publish, or preparation of the manuscript.)

3. We note you have included a table to which you do not refer in the text of your manuscript. Please ensure that you refer to Table 2 in your text; if accepted, production will need this reference to link the reader to the Table.

Reviewers' comments:

Reviewer's Responses to Questions

**Comments to the Author**

1. Is the manuscript technically sound, and do the data support the conclusions?

Reviewer #1: Partly

2. Has the statistical analysis been performed appropriately and rigorously? 

Reviewer #1: Yes

3. Have the authors made all data underlying the findings in their manuscript fully available?

Reviewer #1: No

4. Is the manuscript presented in an intelligible fashion and written in standard English?

Reviewer #1: Yes

5. Review Comments to the Author

Reviewer #1: I have reviewed the manuscript entitled: "nonparametric alternative to the Cochran-Armitage trend test in genetic case-control association studies: The Jonckheere-Terpstra trend test", with Manuscript Number: PONE-D-21-39215. This manuscript is an interesting topic that could help researchers to tackle any bias results might be happened in the related analysis. But still need more revisions (major/minor) that I have mentioned viewpoints/ comments as below:

# Abstract:

Abstract is too general, while it would be better to be more specific in case of the definition and importance of the problem, and authors should give evidences in which are supported with quantitative data such as sample size etc.!

#Introduction:

In Line 47: it seems that sentence is incomplete ...is not additive (e.g.,(Gonzalez et al., 2008;…

Generally: This section at this format seems to be non-informative, authors must give more evidences of reported studies, try to highlight the weakness including suffering from the power loss etc., then discuss importance and objectives of the method of interest with more details.

# Simulation:

In Lines 113~120: authors must present more supportive data in case of sample size and MAFs to show advantages/ dis-advantages of both methods in analyzing genome-wide association studies.

# Real Data Analysis:

At this section, authors are strongly advised to use more real data set for comparing the efficiency of the two tests. With few cases, we are not able to observe power/failure of each method.

# Tables:

In table 2: Could authors show whether the calculated statistics at different sample size are significantly different? By the way, the table was not referred in the main text.

With the best regards.

6. PLOS authors have the option to publish the peer review history of their article (what does this mean?). If published, this will include your full peer review and any attached files.

Reviewer #1: **Yes: **Ali Moumeni

---

## [Author Response · Author response to Decision Letter 0]

19 May 2022

Please see the attached "Plos One response letter.docx" in this resubmission. The content is attached here too:

Point-by-point response to comments from Reviewer

Thank you very much for reviewing our paper and the detailed and helpful review report. We greatly appreciate your time, effort, encouragement, and insight. We have revised our paper addressing all issues raised in your report. The following is our point-by-point response to your comments. For convenience, your original comments are copied and our replies follow in blue. The associated revisions in the manuscript are highlighted in track change.

Comments:

I have reviewed the manuscript entitled: "nonparametric alternative to the Cochran-Armitage trend test in genetic case-control association studies: The Jonckheere-Terpstra trend test", with Manuscript Number: PONE-D-21-39215. This manuscript is an interesting topic that could help researchers to tackle any bias results might be happened in the related analysis. But still need more revisions (major/minor) that I have mentioned viewpoints/ comments as below:

Thank you very much for your concise summary and encouraging comment. We appreciate very much for your time and effort.

# Abstract: 

Abstract is too general, while it would be better to be more specific in case of the definition and importance of the problem, and authors should give evidences in which are supported with quantitative data such as sample size etc.!

Thank you for the comment. In this revision, we have provided more related materials and details in the Abstract. 

#Introduction: 

In Line 47: it seems that sentence is incomplete ...is not additive (e.g.,(Gonzalez et al., 2008;… 

In this sentence, the period is after the long list of citations, so it may appear that the sentence is incomplete:

However, it can suffer from power loss when the true genetic model is not additive (e.g., Gonzalez et al., 2008; Kuo & Feingold, 2010; Li, Zheng, Liang, & Yu, 2009; Loley, König, Hothorn, & Ziegler, 2013). 

When reviewing it, however, we did find an extra “(” after “e.g.,”, and we have removed it in this revision (line 55, page 4).

Generally: This section at this format seems to be non-informative, authors must give more evidences of reported studies, try to highlight the weakness including suffering from the power loss etc., then discuss importance and objectives of the method of interest with more details.

We appreciate this suggestion to add more details in Introduction. In this revision, we expanded Introduction by providing more context information and details for the tests and significance of the research question. 

# Simulation:

In Lines 113~120: authors must present more supportive data in case of sample size and MAFs to show advantages/ dis-advantages of both methods in analyzing genome-wide association studies.

In the original simulation, we considered sample sizes of N∈(200,500,1000) and MAFs of q∈(0.05,0.1,0.2,0.3). To evaluate the relative performance of the CA and JT trend tests in a wider range of data scenarios, we further considered the sample size of N=1500 and 2000 as well as q=0.4. Because the sample sizes and MAF were fairly large, we reduced the effect size in the alternative hypothesis to λ=0.5 to make the power comparison meaningful (otherwise the power of both tests will be close to 1 and hence difficult to evaluate the relative performance). 

The empirical power of the two tests for the additional simulation settings were summarized in Table S1 in the Supplementary Materials. The results were consistent with original conclusion: compared to the CA trend test, the JT trend test is more powerful when the underlying genetic model is dominant. The power advantage of T_JT diminishes as the genetic model evolves toward the additive model and have approximately equivalent power when the underlying model is additive. T_JT becomes less powerful than T_CA^Add and the disadvantage enlarges when the genetic model keeps evolving toward the recessive end.

# Real Data Analysis:

At this section, authors are strongly advised to use more real data set for comparing the efficiency of the two tests. With few cases, we are not able to observe power/failure of each method.

Thank you for the suggestion. In this revision, we compared the two tests on additional studies for SNPs that were reported associated with hypertension. The results were reported in Table S2 in the Supplementary Materials. The conclusion still holds in this real data analysis: T_JT and T_CA^Add had similar power when the genetic model tended to be additive (rs7961152, rs1937506, rs6997709) and T_JT is more powerful than T_CA^Add when the genetic model tended to be dominant (rs2398162).

# Tables:

In table 2: Could authors show whether the calculated statistics at different sample size are significantly different? By the way, the table was not referred in the main text.

Thank you for the comment and we apologize for not referring Table 2 in the main text in the original submission. This table refers to the simulation findings verification using expected theoretical tables for each simulation setting (last paragraph of Simulation section) and we now explained Table 2 in the right place of the manuscript. 

We would like to explain that the test statistics (T_JT and T_CA^Add) in Table 2 were calculated based on “expected theoretical tables”. Specifically, in each simulation setting, using the fixed combination of sample size, MAF, and genetic model, the cell probabilities of each genotype for cases and controls in the genotype distribution table (Table 1) can be calculated, and therefore, the expected cell values can also be calculated by multiplying the probabilities with the sample size. We refer such table consisted of the expected cell values as a “expected” table, which allows us to evaluate the relative performance of the two tests by comparing their theoretical test statistics (T_JT and T_CA^Add) in each simulation setting. The relative difference between theoretical test statistics (ΔT=((T_JT-T_CA^Add ))/(T_CA^Add )×100%) are then reported in Table 2, which are used to verify the empirical findings observed in simulation. To fully explain this, we also provided more details and clarifications to this paragraph in this revision. However, since each simulation setting corresponds to a single ΔT that is non-random, we cannot assess the statistical significance of ΔT. Instead, readers may refer to Figure 1 for the actual power comparison from simulation, which can serve as a proxy for the relative statistical significance between the two tests.

---

## [Editor Report · Decision Letter 1]

23 May 2022

PONE-D-21-39215R1A nonparametric alternative to the Cochran-Armitage trend test in genetic case-control association studies: the Jonckheere-Terpstra trend testPLOS ONE

Dear Dr. Zhou,

Thank you for submitting your manuscript to PLOS ONE. After careful consideration, we feel that it has merit but does not fully meet PLOS ONE’s publication criteria as it currently stands. Therefore, we invite you to submit a revised version of the manuscript that addresses the points raised during the review process.

We look forward to receiving your revised manuscript.

Kind regards,

Mehdi Rahimi, Ph.D.

Academic Editor

PLOS ONE

Additional Editor Comments:

Dear Author

The reviewer(s) have recommended major revisions to your manuscript. Therefore, I invite you to respond to the reviewer(s)' comments and revise your manuscript.

With Thanks
---

## [Author Response · Author response to Decision Letter 1]

26 Aug 2022

Point-by-point response to comments from Reviewer

Thank you very much for reviewing our paper and the detailed and helpful review report. We greatly appreciate your time, effort, encouragement, and insight. We have revised our paper addressing all issues raised in your report. The following is our point-by-point response to your comments. For convenience, your original comments are copied and our replies follow in blue. The associated revisions in the manuscript are highlighted in track change.

Comments:

I have reviewed the manuscript entitled: "nonparametric alternative to the Cochran-Armitage trend test in genetic case-control association studies: The Jonckheere-Terpstra trend test", with Manuscript Number: PONE-D-21-39215. This manuscript is an interesting topic that could help researchers to tackle any bias results might be happened in the related analysis. But still need more revisions (major/minor) that I have mentioned viewpoints/ comments as below:

Thank you very much for your concise summary and encouraging comment. We appreciate very much for your time and effort.

# Abstract: 

Abstract is too general, while it would be better to be more specific in case of the definition and importance of the problem, and authors should give evidences in which are supported with quantitative data such as sample size etc.!

Thank you for the comment. In this revision, we have provided more related materials and details in the Abstract. 

#Introduction: 

In Line 47: it seems that sentence is incomplete ...is not additive (e.g.,(Gonzalez et al., 2008;… 

In this sentence, the period is after the long list of citations, so it may appear that the sentence is incomplete:

However, it can suffer from power loss when the true genetic model is not additive (e.g., Gonzalez et al., 2008; Kuo & Feingold, 2010; Li, Zheng, Liang, & Yu, 2009; Loley, König, Hothorn, & Ziegler, 2013). 

When reviewing it, however, we did find an extra “(” after “e.g.,”, and we have removed it in this revision (line 55, page 4).

Generally: This section at this format seems to be non-informative, authors must give more evidences of reported studies, try to highlight the weakness including suffering from the power loss etc., then discuss importance and objectives of the method of interest with more details.

We appreciate this suggestion to add more details in Introduction. In this revision, we expanded Introduction by providing more context information and details for the tests and significance of the research question. 

# Simulation:

In Lines 113~120: authors must present more supportive data in case of sample size and MAFs to show advantages/ dis-advantages of both methods in analyzing genome-wide association studies.

In the original simulation, we considered sample sizes of N∈(200,500,1000) and MAFs of q∈(0.05,0.1,0.2,0.3). To evaluate the relative performance of the CA and JT trend tests in a wider range of data scenarios, we further considered the sample size of N=1500 and 2000 as well as q=0.4. Because the sample sizes and MAF were fairly large, we reduced the effect size in the alternative hypothesis to λ=0.5 to make the power comparison meaningful (otherwise the power of both tests will be close to 1 and hence difficult to evaluate the relative performance). 

The empirical power of the two tests for the additional simulation settings were summarized in Table S1 in the Supplementary Materials. The results were consistent with original conclusion: compared to the CA trend test, the JT trend test is more powerful when the underlying genetic model is dominant. The power advantage of T_JT diminishes as the genetic model evolves toward the additive model and have approximately equivalent power when the underlying model is additive. T_JT becomes less powerful than T_CA^Add and the disadvantage enlarges when the genetic model keeps evolving toward the recessive end.

# Real Data Analysis:

At this section, authors are strongly advised to use more real data set for comparing the efficiency of the two tests. With few cases, we are not able to observe power/failure of each method.

Thank you for the suggestion. In this revision, we compared the two tests on additional studies for SNPs that were reported associated with hypertension. The results were reported in Table S2 in the Supplementary Materials. The conclusion still holds in this real data analysis: T_JT and T_CA^Add had similar power when the genetic model tended to be additive (rs7961152, rs1937506, rs6997709) and T_JT is more powerful than T_CA^Add when the genetic model tended to be dominant (rs2398162).

# Tables:

In table 2: Could authors show whether the calculated statistics at different sample size are significantly different? By the way, the table was not referred in the main text.

Thank you for the comment and we apologize for not referring Table 2 in the main text in the original submission. This table refers to the simulation findings verification using expected theoretical tables for each simulation setting (last paragraph of Simulation section) and we now explained Table 2 in the right place of the manuscript. 

We would like to explain that the test statistics (T_JT and T_CA^Add) in Table 2 were calculated based on “expected theoretical tables”. Specifically, in each simulation setting, using the fixed combination of sample size, MAF, and genetic model, the cell probabilities of each genotype for cases and controls in the genotype distribution table (Table 1) can be calculated, and therefore, the expected cell values can also be calculated by multiplying the probabilities with the sample size. We refer such table consisted of the expected cell values as a “expected” table, which allows us to evaluate the relative performance of the two tests by comparing their theoretical test statistics (T_JT and T_CA^Add) in each simulation setting. The relative difference between theoretical test statistics (ΔT=((T_JT-T_CA^Add ))/(T_CA^Add )×100%) are then reported in Table 2, which are used to verify the empirical findings observed in simulation. To fully explain this, we also provided more details and clarifications to this paragraph in this revision. However, since each simulation setting corresponds to a single ΔT that is non-random, we cannot assess the statistical significance of ΔT. Instead, readers may refer to Figure 1 for the actual power comparison from simulation, which can serve as a proxy for the relative statistical significance between the two tests.

---

## [Decision Letter · Decision Letter 2]

24 Oct 2022

PONE-D-21-39215R2A nonparametric alternative to the Cochran-Armitage trend test in genetic case-control association studies: the Jonckheere-Terpstra trend testPLOS ONE

Dear Dr. Zhou,

Thank you for submitting your manuscript to PLOS ONE. After careful consideration, we feel that it has merit but does not fully meet PLOS ONE’s publication criteria as it currently stands. Therefore, we invite you to submit a revised version of the manuscript that addresses the points raised during the review process.

We look forward to receiving your revised manuscript.

Kind regards,

Mehdi Rahimi, Ph.D.

Academic Editor

PLOS ONE

Journal Requirements:

Reviewers' comments:

Reviewer's Responses to Questions

**Comments to the Author**

1. If the authors have adequately addressed your comments raised in a previous round of review and you feel that this manuscript is now acceptable for publication, you may indicate that here to bypass the “Comments to the Author” section, enter your conflict of interest statement in the “Confidential to Editor” section, and submit your "Accept" recommendation.

Reviewer #2: (No Response)

2. Is the manuscript technically sound, and do the data support the conclusions?

Reviewer #2: No

3. Has the statistical analysis been performed appropriately and rigorously? 

Reviewer #2: No

4. Have the authors made all data underlying the findings in their manuscript fully available?

Reviewer #2: Yes

5. Is the manuscript presented in an intelligible fashion and written in standard English?

Reviewer #2: Yes

6. Review Comments to the Author

Reviewer #2: This article concerns a power comparison of the Cochran-Armitage trend test and the non-parametric Jonckheere-Terpstra trend test, under common genetic models (additive, dominant, recessive), different minor allele frequencies and sample sizes, and for bi-allelic genetic variants. The article is concise and well written, and the conclusions are clear. I have some major and minor concerns detailed below.

Major concerns:

The example in the Real data analysis section on page 9 is not clearly a case of dominance; in fact, a co-dominant model seems to be the best, where the heterozygote has the highest risk. Variant rs20541 is thus a poor example to make the case of an advantage of the Jonckheere-Terpstra test. In fact, the clearest case of dominance is rs2398162 in Table S2. If the authors wish to practically illustrate the advantage of the Jonckheere-Terpstra test, then rs2398162 would be a better choice.

On line 192 the authors state “variant rs10900589 in ATP2B4 was associated with the disease in the Ghanaian samples and the association was replicated in the Gambian samples”. This statement is obviously FALSE. For Figure 2 shows very significant association for the Ghanaian sample, but a clearly non-significant association for the Gambian sample. Please correct the sentence.

In genetic association studies it is common to test the SNPs involved for Hardy-Weinberg equilibrium. If there are significant deviations from HWE, the results of the association tests may be questioned, because disequilibrium is potentially indicative of the presence of genotyping errors (Hosking et al, 2004; Leal, 2005). The HWE testing can be done with many genetic data analysis software such as PLINK (Purcell et al., 2007) or R-package Hardy-Weinberg (Graffelman, 2015). It is recommended to test cases and controls separately, and report exact Hardy-Weinberg p-values for each group. Exact testing is the preferred approach, for it has the highest power. I suggest the authors to include HW test results in the paper, for this can only improve the credibility of their conclusions.

The authors emphasize the power gain of the Jonckheere-Terpstra test under the dominant model. However, the power loss of this test under recessive model is generally much larger than the gain under the dominant model (see Figure 1, Table 2). In practice one often does not know the correct genetic model. One thus say that, a priori, and overall, the Cochran Armitage test may be the best choice, at least if the MAF is not low. This conclusion should be added to the evaluation of the tests in the Discussion section.

In genetic association studies with SNPs, it is also common to test for association not only at the genotype level, but also at the level of alleles. A standard test for this purpose is the so called alleles test (Laird & Lange, 2011). For all empirical 9 SNPs (Page Figure 2, Table S2) the alleles test leads to the same conclusion as the Jonckheere-Terpstra test. This could at least be stated, as it strengthens the conclusions of the “Real data analysis” Section.

Minor issues:

L59: “recource” -- “resource”

L138: “set of simulation” -- “set of simulations”

L161: “We refer this table consisted of” -- “This table consists of”

L162: delete “as an expected table”.

L167, L202: “were reported” -- “are reported”

L262: capitalize “Kendall”

References:

Graffelman, J. (2015) Exploring Diallelic Genetic Markers: The HardyWeinberg Package. The Journal of Statistical Software 64(3): 1–23.

Hosking, L., S. Lumsden, K. Lewis, A. Yeo, L. McCarthy, A. Bansal, J. Riley, I. Purvis, and C. Xu (2004): Detection of genotyping errors by Hardy-Weinberg equilibrium testing. Eur. J. Hum. Genet., 12, 395–399.

Laird, N. M. and Lange, C. (2011) The fundamentals of modern statistical genetics. Springer.

Leal SM (2005) Detection of genotyping errors and pseudo-SNPs via deviations from Hardy–Weinberg equilibrium. Genet Epidemiol 29:204–214

S. Purcell and B. Neale and K. Todd-Brown and L. Thomas and M. A. R. Ferreira and D. Bender and J. Maller and P. Sklar and P. I. W. de Bakker and M. J. Daly and P. C. Sham (2007) PLINK: A Toolset for Whole-Genome Association and Population-Based Linkage Analysis. American Journal of Human Genetics 81(3): 559—575.

7. PLOS authors have the option to publish the peer review history of their article (what does this mean?). If published, this will include your full peer review and any attached files.

Reviewer #2: No

---

## [Author Response · Author response to Decision Letter 2]

16 Dec 2022

The word document of response letter is included in this submission too.

Reviewer #2: This article concerns a power comparison of the Cochran-Armitage trend test and the non-parametric Jonckheere-Terpstra trend test, under common genetic models (additive, dominant, recessive), different minor allele frequencies and sample sizes, and for bi-allelic genetic variants. The article is concise and well written, and the conclusions are clear. I have some major and minor concerns detailed below.

Thank you very much for your concise summary and encouraging comment. We appreciate very much for your time and effort.

Major concerns: 

The example in the Real data analysis section on page 9 is not clearly a case of dominance; in fact, a co-dominant model seems to be the best, where the heterozygote has the highest risk. Variant rs20541 is thus a poor example to make the case of an advantage of the Jonckheere-Terpstra test. In fact, the clearest case of dominance is rs2398162 in Table S2. If the authors wish to practically illustrate the advantage of the Jonckheere-Terpstra test, then rs2398162 would be a better choice.

Thank you for this suggestion. In this revision we updated this section by using the variant rs2398162 in the hypertension study in the Real data analysis as an example of dominant model. Relevant text and Table S2 were also updated to reflect this change.

On line 192 the authors state “variant rs10900589 in ATP2B4 was associated with the disease in the Ghanaian samples and the association was replicated in the Gambian samples”. This statement is obviously FALSE. For Figure 2 shows very significant association for the Ghanaian sample, but a clearly non-significant association for the Gambian sample. Please correct the sentence.

Thank you for this observation and we apologize that the previous statement was inaccurate. In Loley et al. (2013, EJHG 21:1442-1448) it was shown this signal was only significant when coded in a recessive model in the Gambian group (Table 1). In this revision we corrected the sentence and stated that “This association was also evaluated in the Gambian samples and it was significant under a recessive model but insignificant under dominant and additive models.”

In genetic association studies it is common to test the SNPs involved for Hardy-Weinberg equilibrium. If there are significant deviations from HWE, the results of the association tests may be questioned, because disequilibrium is potentially indicative of the presence of genotyping errors (Hosking et al, 2004; Leal, 2005). The HWE testing can be done with many genetic data analysis software such as PLINK (Purcell et al., 2007) or R-package Hardy-Weinberg (Graffelman, 2015). It is recommended to test cases and controls separately, and report exact Hardy-Weinberg p-values for each group. Exact testing is the preferred approach, for it has the highest power. I suggest the authors to include HW test results in the paper, for this can only improve the credibility of their conclusions.

Thank you for this suggestion. In the revision we conducted the exact test for HWE among the cases and controls for each of the variant using the suggested R package (HardyWeinberg). The p-value results were summarized in S3 Table and we concluded that “Results showed that the p-values of the HWE tests for all the variants were larger than 0.01, with only two between 0.01 and 0.05, suggesting that there was little evidence of genotyping error among the variants”.

The authors emphasize the power gain of the Jonckheere-Terpstra test under the dominant model. However, the power loss of this test under recessive model is generally much larger than the gain under the dominant model (see Figure 1, Table 2). In practice one often does not know the correct genetic model. One thus say that, a priori, and overall, the Cochran Armitage test may be the best choice, at least if the MAF is not low. This conclusion should be added to the evaluation of the tests in the Discussion section

Thank you for bringing up this excellent point. In this revision, we provided such information about the selection between JT and CA trend tests when the true genetic model is unknown at the end of the discussion.

In genetic association studies with SNPs, it is also common to test for association not only at the genotype level, but also at the level of alleles. A standard test for this purpose is the so called alleles test (Laird & Lange, 2011). For all empirical 9 SNPs (Page Figure 2, Table S2) the alleles test leads to the same conclusion as the Jonckheere-Terpstra test. This could at least be stated, as it strengthens the conclusions of the “Real data analysis” Section.

Thank you for this suggestion. We conducted the allelic test for the variants included in the Real data analysis section and confirmed that the test results were close to the discussed methods. The results of the allelic test were included in S3 table. 

Minor issues: 

L59: “recource” -- “resource” 

L138: “set of simulation” -- “set of simulations” 

L161: “We refer this table consisted of” -- “This table consists of” 

L162: delete “as an expected table”. 

L167, L202: “were reported” -- “are reported” 

L262: capitalize “Kendall” 

Thank you for these observations. We have made corresponding changes in this revision.

---

## [Decision Letter · Decision Letter 3]

10 Jan 2023

A nonparametric alternative to the Cochran-Armitage trend test in genetic case-control association studies: the Jonckheere-Terpstra trend test

PONE-D-21-39215R3

Dear Dr. Zhou,

We’re pleased to inform you that your manuscript has been judged scientifically suitable for publication and will be formally accepted for publication once it meets all outstanding technical requirements.

Kind regards,

Mehdi Rahimi, Ph.D.

Academic Editor

PLOS ONE

Additional Editor Comments (optional):

Reviewers' comments:

Reviewer's Responses to Questions

**Comments to the Author**

1. If the authors have adequately addressed your comments raised in a previous round of review and you feel that this manuscript is now acceptable for publication, you may indicate that here to bypass the “Comments to the Author” section, enter your conflict of interest statement in the “Confidential to Editor” section, and submit your "Accept" recommendation.

Reviewer #2: All comments have been addressed

2. Is the manuscript technically sound, and do the data support the conclusions?

Reviewer #2: Yes

3. Has the statistical analysis been performed appropriately and rigorously? 

Reviewer #2: Yes

4. Have the authors made all data underlying the findings in their manuscript fully available?

Reviewer #2: Yes

5. Is the manuscript presented in an intelligible fashion and written in standard English?

Reviewer #2: Yes

6. Review Comments to the Author

Reviewer #2: This article concerns a power comparison of the Cochran-Armitage trend test and the non-parametric Jonckheere-Terpstra trend test, under common genetic models (additive, dominant, recessive), different minor allele frequencies and sample sizes, and for bi-allelic genetic variants. The article has improved after the previous round of review. The authors have addressed most of my concerns satisfactorily. I have only some minor points for improvement left which are detailed below.

L75: “as Table” --- “as in Table”

L82: The null hypothesis is supposed to apply to all i (0, 1 or 2); that should be stated.

7. PLOS authors have the option to publish the peer review history of their article (what does this mean?). If published, this will include your full peer review and any attached files.

Reviewer #2: No

---

## [Editor Report · Acceptance letter]

24 Jan 2023

PONE-D-21-39215R3 

A nonparametric alternative to the Cochran-Armitage trend test in genetic case-control association studies: the Jonckheere-Terpstra trend test 

Dear Dr. Zhou:

I'm pleased to inform you that your manuscript has been deemed suitable for publication in PLOS ONE. Congratulations! Your manuscript is now with our production department. 

Kind regards, 

on behalf of

Associate Prof. Mehdi Rahimi 

Academic Editor

PLOS ONE